# Preparation of a Nickel Layer with Bell-Mouthed Macropores via the Dual-Template Method

**Ruishan Yang, Weiguo Yao, Guangguang Qian and Yanli Dou \***

The Ministry of Education Key Laboratory of Automotive Material, College of Material Science and Engineering, Jilin University, Changchun 130015, China; yangrs18@mails.jlu.edu.cn (R.Y.); wgyao@jlu.edu.cn (W.Y.); qiangg19@mails.jlu.edu.cn (G.Q.)
\* Correspondence: douyl@jlu.edu.cn

**Abstract:** A relatively static and unique bubble template is successfully realized on a microporous substrate by controlling the surface tensions of the electrodeposit solution, and a nickel layer containing macropores is prepared using this bubble template. When the surface tension of the solution is 50.2 mN/m, the desired bubble template can be formed, there are fewer bubbles attached to other areas on the substrate, and a good nickel layer is obtained. In the analysis of the macropore formation process, it is found that the size of the bell-mouthed macropores can be tailored by changing the solution stirring speed or the current density to adjust the growth rate of the bubble template. The size change of a macropore is measured by the profile angle of the longitudinal macropore, section. As the solution stirring speed increases from 160 to 480 r/min, the angle range of the bell-mouthed macropores cross-sectional profile is increased from $21.0°$ to $44.3°$. In addition, the angle range of the bell-mouthed macropore cross-sectional profile is increased from $39.3°$ to $46.3°$ with the current density increasing from 1 to 2.5 A/dm$^2$. Different from the dynamic hydrogen bubble template, the bubble template implemented in this paper stays attached on the deposition and grows slowly, which is novel and interesting, and the nickel layer containing macropores prepared using this bubble template is applied in completely different fields.

**Keywords:** bell-mouthed macropore; bubble template; relatively static; electrodeposition

## 1. Introduction

The dynamic hydrogen bubble template (DHBT) method has been widely studied in recent years, and the materials with high specific surface areas prepared by this method have important applications for catalysis, sensing, and energy [1–4]. In the DHBT method, driven by the high driving force conditions of high over-potential [5] and sufficient proton sources [6,7], dense hydrogen-evolution-active sites form on the surface of the cathode and bubbles are generated and desorbed quickly. The metal ion is deposited with this dynamic bubble template [8], and then, a micron-thickness metal layer with microporous and nanoramified metal walls is prepared efficiently in a few to tens of seconds [9,10]. The morphology of the deposition is greatly affected by the bubble behavior, which basically involves three basic steps, nucleation, growth, and detachment [11,12], and many factors affect the steps, such as the type and concentration of metal ions [13,14], additives [15,16], current density [5], and electrode material [17]. Generally, an effective bubble template can be produced to make the deposition porous with a sufficient proton source in solution [13]. When the bubbles appear for a longer residence time [17,18], larger pores in the deposition are obtained, and smaller pores form in the deposition when the coalescence of bubbles is suppressed [15,19–21]. It is worth noting that the bubbles will coalesce during the evolution process, resulting in the pore size of the deposition increasing with increasing distance from the substrate [19,22,23]. For this phenomenon, the bubble template behavior proposed by Liu [22] is typical, in which the bubbles become increasingly larger during the bubble evolution. Although the bubbles in the DHBT method evolve rapidly and intensely,

by adjusting the deposition parameters to control the bubble behavior, the morphology of the deposition can be controllable to some degree. To date, however, most studies have focused just on the preparation of the microporous metal deposits in a short electric plating time using the DHBT method. The potential applications of this method to make macropores with smooth walls in the metal deposits over a long electric plating time are yet to be explored.

There are some reports of the electrodeposition of a smooth nickel coating indicating that when the content of the wetting agent in the solution is too small [24] or the substrate surface contains defects [25,26], hydrogen bubbles are liable to attach to the surface of the deposition and form pinhole defects. The pinhole defect distributing independently has a small starting point and a large end. It indicates that the bubbles on the cathode stay attached and gradually grow, and the behavior of this bubble template is relatively static and completely different from that of the DHBT, where bubbles are generated and desorbed quickly. It is noticeable that this relatively static bubble template is created under electrodeposition conditions involving a buffer [27] and a lower current density [28,29] to suppress the hydrogen evolution reaction. All these indicate that the appearance of this bubble template is induced by a certain deposition condition and surface structure of the substrate, and it is conceivable that if the deposition conditions are controlled accurately, the generation of the bubble template may be regularized. The position of the bubble template created in this way is controllable, which is superior to the characteristic that bubbles are randomly generated under a high driving force in the DHBT. Thus, a controllable morphology nickel layer with bell-mouthed macropores is realized. To the best of our knowledge, till now, there have not been any reports on the use of the defects on the substrate and the relatively static bubble template to prepare a nickel layer with large pores.

In this study, a copper sheet with micropores is used as the first template in the electrodeposition of nickel. The hydrogen bubbles take these micropores as the starting point to grow by merging with micro-bubbles generated on the surface of the deposition. These growing bubbles with fixed positions act as the second template of the nickel deposition. The optimal composition of the solution to form the desired bell-mouthed macropores in the nickel deposition is investigated. Through the analysis of the desired bell-mouthed macropore formation mechanism, the factors affecting the bell-mouthed macropore morphology were discovered and explored. A controllable-morphology nickel layer with millimeter thickness and millimeter-sized bell-mouthed macropores was formed through electrodeposition for tens of hours. The nickel layer can be used to prepare a mold structure in which the mold cavity can be vacuumed through the bell-mouthed macropores distributed in the mold to add the vacuum forming function to the traditional electroforming mold.

## 2. Materials and Methods

A $100 \times 65 \times 0.2$ mm$^3$ Hull copper sheet is used as the cathode substrate, which is insulated with a plastic film on one side. Micropores of 200 μm are fabricated by laser on the copper sheet, and then a drop insulating glue (Kafuter) is applied on the opening of the micropore on the insulating side of the copper sheet to block the micropore. Before electrodeposition, the copper sheet is immersed in 5%wt HCl solution for 10 min, then rinsed with deionized water, and subsequently dried in hot air. The process from substrate preparation to electrodeposition is shown in Figure 1. For the purpose of this experiment, the substrate is prepared by the above method, and there is no reference preparation standard. The plum blossom nickel (99.8% purity, Inco) is used as the anode, and it is necessary to wrap the anode in a polyester bag because of a long deposition time.

A mixed solution of 380 g/L Ni(NH$_2$SO$_3$)$_2$·4H$_2$O (99.8% purity, Sinopharm Chemical Reagent Co., Ltd., Shanghai, China), 5 g/L NiCl$_2$·6H$_2$O (99.8% purity, Sinopharm Chemical Reagent Co., Ltd., Shanghai, China), and 36 g/L H$_3$BO$_3$ (99.8% purity, Sinopharm Chemical Reagent Co., Ltd., Shanghai, China) is used as the base bath. The content of sodium dodecyl

sulfate (SDS, 99.8% purity, Xilong Scientific Co., Ltd., Shantou, China) is 0.0020–0.0100 g/L in the bath. Nickel deposition is obtained on applying a 1.5 A/dm$^2$ cathodic current density at 50 °C for 24 h in a 1000 mL beaker. The beaker is placed in a water bath pot (Yuhua Instrument Company, Zhengzhou, China), which provides heating and stirring for the electrodeposition. A Direct currentpower supply (Longwei Instrument Co., Ltd., Hongkong, China) is used to provide power.

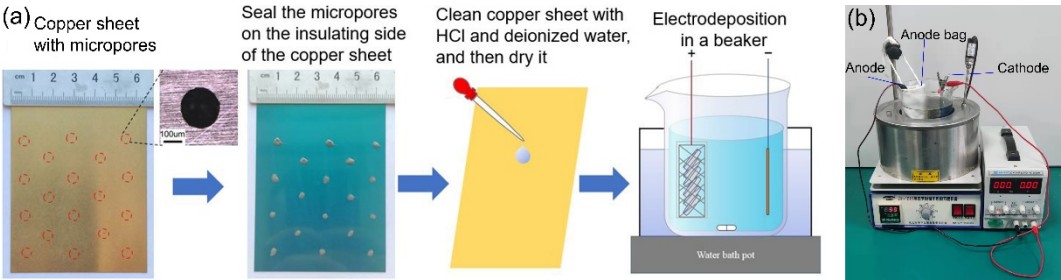

**Figure 1.** (**a**) Substrate preparation process; (**b**) experimental equipment.

The solution with different contents of SDS is heated to 50 °C, and then the surface tension of the solution is quickly tested using a surface tensiometer (Youte Testing Instrument Manufacturing Co., Ltd., Chengde, China). The nickel deposition is cut by a cutting machine (Dongcheng Power Tools Co., Ltd., Nantong, China) in the vertical direction; then a file is used to grind the section to the symmetry line of the macropore; and then 400- to 1200-grit sandpaper is used to grind the section to obtain the section morphology. A digital camera (Canon, Tokyo, Japan) is used to directly observe the surface morphology of the deposition, and the section morphology of the bell-mouthed macropores are collected by an optical microscope (CNOPTEC, Chongqing, China). In the software Photoshop, the outline of the section morphology of the macropore is extended to intersect and the measured angle is used to measure the size of the macropore.

## 3. Results and Discussion

### 3.1. Analysis of the Formation of Bell-Mouthed Macropores in Deposition

SDS is a common surfactant, which can effectively reduce the surface tension of the solution, leading to reduced adhesion energy to promote the desorption of bubbles from the cathode surface. The morphology of the nickel layers prepared in the electrolyte with different contents of SDS is directly observed by digital photos. As shown in Figure 2, the contents of SDS in Figure 2a,b are 0.0020 and 0.0040 g/L and there are a large number of pores on the deposition, in which the pores encircled by dotted red lines are desired and the rest are pinhole defects. With the increase in the SDS content, the number of pinholes decreases in Figure 2a–f. This is because the adhesion area of the bubbles attached to the cathode surface is small, because of which the bubbles are easy to desorb as the surface tension of the solution decreases. However, the adhesion interface of the bubbles attached to the micropores in the micropore section is larger than the adhesion interface of the bubbles attached to the smooth surface, so the bubbles attached to the micropores are difficult to desorb.

**Table 1.** Surface tension of the solution with different contents of sodium lauryl sulfate.

| SDS (g/L) | 0.0020 | 0.0040 | 0.0050 | 0.0060 | 0.0080 | 0.0100 |
|---|---|---|---|---|---|---|
| Surface tension (mN/m) | 60.0 ± 0.3 | 53.4 ± 0.2 | 50.2 ± 0.2 | 47.4 ± 0.3 | 46.7 ± 0.1 | 46.0 ± 0.1 |

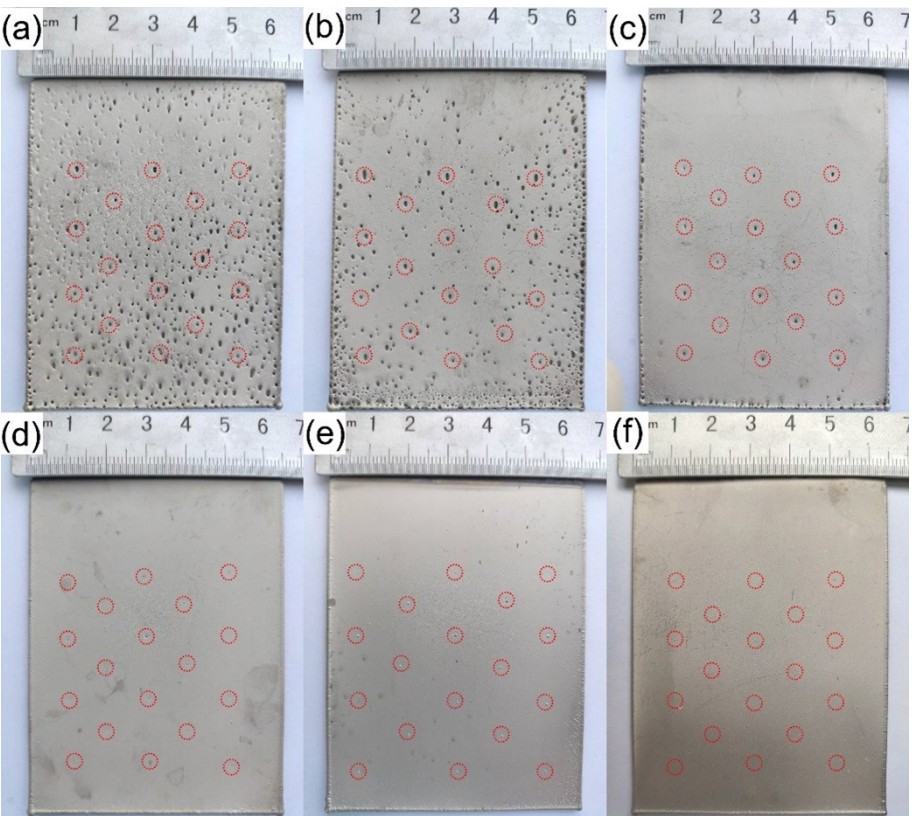

**Figure 2.** Digital photo of the electrodeposition sample produced as Table 1. Content of SDS (g/L):
(**a**) 0.0020 g/L; (**b**) 0.0040 g/L; (**c**) 0.0050 g/L; (**d**) 0.0060 g/L; (**e**) 0.0080 g/L; (**f**) 0.0100 g/L.

Equation (1) can be used to express the relationship between unit adhesion energy
($w_{ad}$), the surface tension of the H$_2$–liquid interface ($\gamma_L$), and the contact angle of the bubble
on the cathode surface ($\theta_{H2}$) [15,16].

$$w_{ad} = \frac{E_{ad}}{S_{air}} = \gamma_L(1 + cos\theta_{H2}), \tag{1}$$

where $E_{ad}$ is the adhesion energy, which is defined as the work required in detaching the
bubble from the electrode, and $S_{air}$ is the area of the interface. If the surface tension of the
solution is constant, desorption occurs in the bubbles with a smaller adhesion interface or
the contact angle between the bubbles and the cathode surface gradually increases as the
bubbles grow [30].

As shown in Figure 2c, the pinholes in the smooth region of the substrate have
disappeared, while bell-mouthed macropores are formed in the deposition at the positions
of micropores on the substrate. Therefore, a position-controllable bubble template is
realized. Figure 3a shows a section morphology of the bell-mouthed macropores in the
deposition in Figure 2c. However, the desired bell-mouthed macropores have not formed
in the deposition in Figure 2d–f, where the content of SDS is 0.0060, 0.0080, and 0.0100 g/L,
respectively. Some pointed protrusions (Figure 3b) appear on the bonding surface of the
nickel deposition in Figure 2d, which correspond to the position of the micropores made
on the substrate. This protrusion indicates that the solution can wet the micropores on
the substrate and the nickel ions deposited in the micropores, due to which the solution
surface tension is less than 47.4 mN/m (Table 1). Therefore, the optimal composition of the
solution is the base bath with 0.0050 g/L SDS ($\gamma_L$ = 50.2 mN/m).

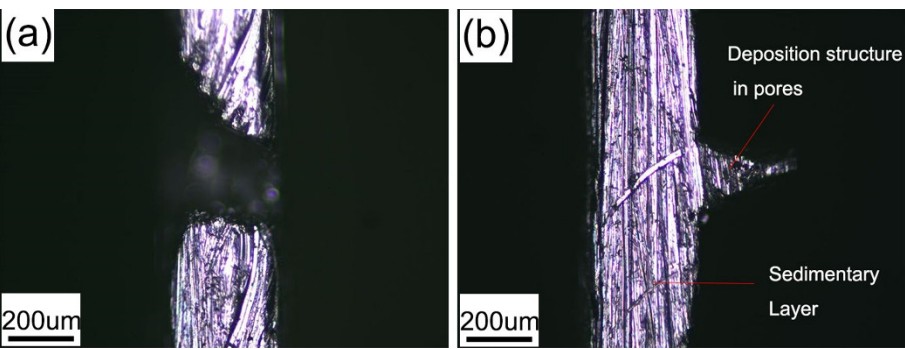

**Figure 3.** (**a**) The section morphology of the bell-mouthed macropores in the deposition in Figure 2c and (**b**) the protrusion structure of the deposition in Figure 2d.

According to the experimental phenomenon, the formation process of desired bell-mouthed macropores in the deposition is speculated and shown in Figure 4. When the copper sheet is immersed in the solution, the solution with a high surface tension cannot wet the micropores and a part of the gas remains in the micropores (Figure 4b). The hydrogen micro-bubbles generated on the surface of the cathode detach and dissolve in the solution [31], and some of them are merged with the gas in the micropores [8] (Figure 4c) as the solution flows through the surface of the gas, so the gas volume in the micropore gradually grows to be bubbles, as shown in Figure 4d. However, nickel ions are continuously deposited on the substrate and this process makes the deposition "climb" along the bubbles, which reduces the buoyancy of the bubbles and provides an additional adhesion area to the bubbles, and both of these effects help the bubbles stay attached to the substrate rather than desorb. It can be seen that unlike the dynamic hydrogen bubble template, the bubble template in Figure 4 is relatively static and can remain attached all the time to provide a steady template for metal deposition. Because the deposition is along the surface of the bubble template, when the contact angle between the bubbles and the surface of the deposition is an obtuse angle, a bell-mouthed macropore will be formed in the deposition. In addition, due to the buoyancy of the bubbles, the bell-mouthed macropores in the deposition are inclined upward. It should be noted that the size of the micropores on the substrate is an important parameter that affects the formation of the bubble template. In the previous discussion about the attachment of bubbles onto the surface of the substrate, the size of the micropores was larger than the attachment interface of the bubbles on the flat area, which is conducive to the adhesion of the bubbles at the micropores. However, according to the explanation of the bubble template formation process, it can be reasonably assumed that when the diameter of the micropore is too large, the solution will wet the inside of the micropores and the desired bubble template will not form. Therefore, the effect of micropore size on the formation of the bubble template is complicated.

The size of the bell-mouthed macropores in the deposition is determined by the size of the bubbles. Thus, size-controllable bell-mouthed macropores can be obtained by controlling the growth rate of the bubbles. Actually, the growth rate of the bubbles on the micropores depends on the rate at which they merge with the micro-bubbles in the solution. The merge rate is related to the concentration of hydrogen micro-bubbles in the solution and the flow velocity of the electrolyte on the cathode surface. The concentration of hydrogen micro-bubbles in the solution will increase when the current density increases. The flow velocity of the electrolyte on the cathode surface is directly determined by the stirring speed of the solution. Therefore, the effect of flow velocity and current density on the morphology of bell-mouthed macropores in the deposition is discussed below.

*3.2. Effect of Stirring Speed of the Solution*

Stirring of the solution during electrodeposition is necessary to promote mass transfer. The solution stirring speed is set to 240, 320, 400, and 480 r/min in the base bath containing

0.005 g/L SDS, and deposition occurs at a current density of 1.5 A/dm$^2$ for 120 h so the influence of the stirring speed on the size of the bell-mouthed macropores in the deposition can be observed. Figure 5a–d shows the frontal morphology of the depositions at different stirring speeds. The edges of the desired bell-mouthed macropores circled by dotted red lines are elliptical, which is because the attached bubbles move upward under buoyancy. In addition to the desired bell-mouthed macropores in the deposition, there are some pinhole defects. This may be because the SDS is consumed during the electrodeposition process, which makes it easy for the bubbles to adhere to the cathode surface.

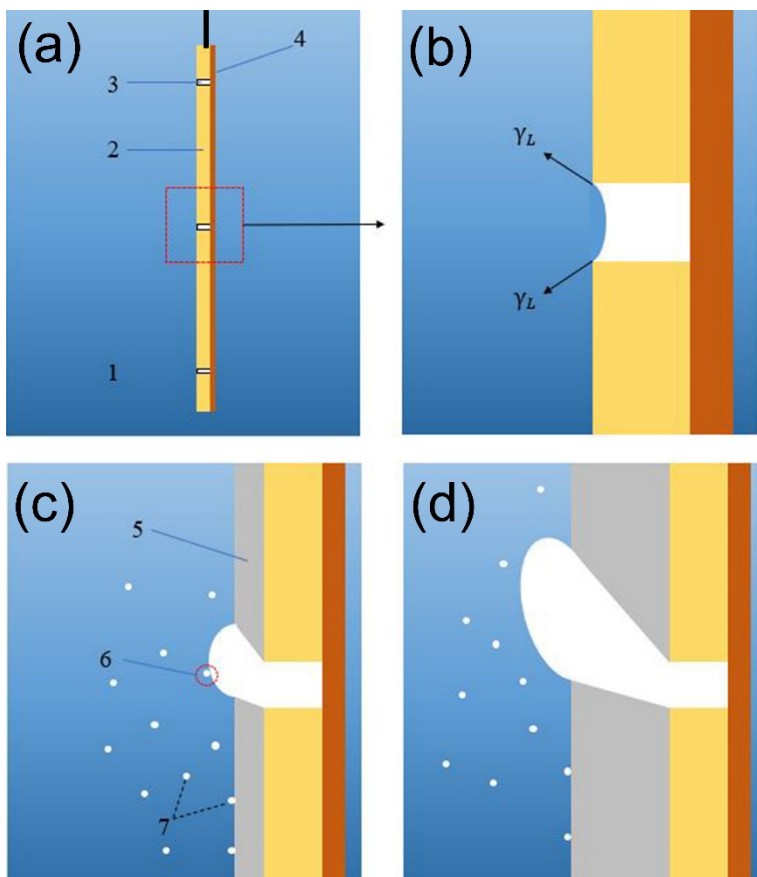

**Figure 4.** The formation process of bell-mouthed macropores in the deposition: (**a**) the substrate is placed in the solution; (**b**) state of micropores before electrodeposition; (**c**) electrodeposition begins and macro-bubbles are formed; (**d**) bubbles and deposition in electrodeposition. (1, bath solution; 2, Hull copper sheet; 3, micropore on the copper sheet; 4, insulating layer; 5, nickel deposition; 6, micro-bubbles merging with bubbles on the cathode; 7, micro-bubbles generated on the cathode and dissolved in the solution).

Cut the bell-mouthed macropores along the vertical symmetry axis of the macropore edge to observe their cross-sectional morphology. The cross section of the bell-mouthed macropores obtained in this way is the largest and can accurately reflect the evolution of the entire macropore. Figure 5(a1–a4,b1–b4,c1–c4,d1–d4) show the morphologies of bell-mouthed macropores at stirring speeds of 240, 320, 400, and 480 r/min, respectively. It can be seen that the bell-mouthed macropores from a to d increase significantly. In Figure 5, the extended contour line measures the opening angle of the bell-mouthed macropores, and the angle values are listed in Table 2. As the solution stirring speed increases from 160 to 480 r/min, the angle range of the bell-mouthed macropore cross-sectional profile is increased from 21.0° to 44.3°. Although the angle ranges under different current densities partially overlap, in general, the bell-mouthed macropores become larger with an increase

in the current density. Because the increase in stirring speed makes the flow velocity of the electrolyte on the cathode surface increase, the bubbles attached to the bell-mouthed macropores capture more hydrogen micro-bubbles dissolved in the solution, thereby accelerating their growth. Thus, the bell-mouthed macropores become larger.

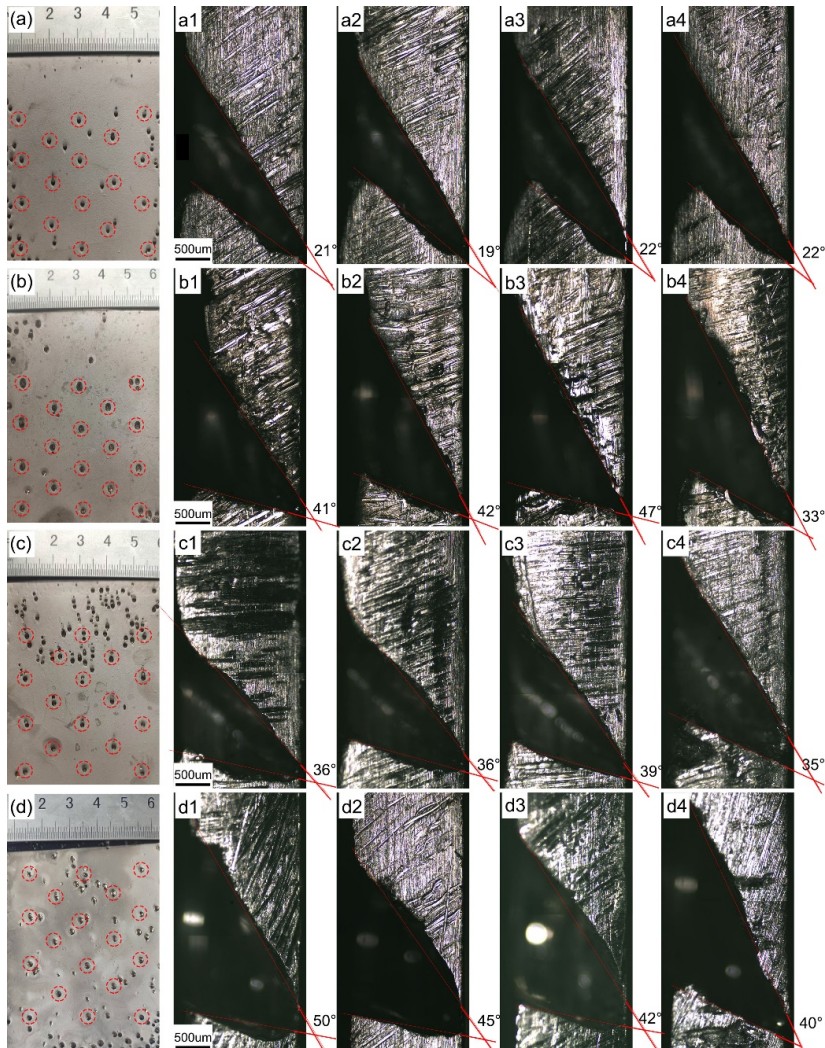

**Figure 5.** The morphology of depositions at different stirring speeds: (**a**) 240 r/min; (**b**) 320 r/min; (**c**) 400 r/min; and (**d**) 480 r/min. (**a**–**d**) and (**a1**–**d4**) are the front and cross-sectional morphology of the macropore, respectively.

**Table 2.** Angles of bell-mouthed macropores' outline in depositions at different stirring speeds.

| Stirring speed (r/min) | 240 | 320 | 400 | 480 |
|---|---|---|---|---|
| Average angle of macropores profile (°) | 21.0 ± 1.4 | 39.5 ± 4.3 | 36.5 ± 1.7 | 44.3 ± 4.3 |

### 3.3. Effect of Current Density

In the following experiment, the current density is set to 1, 1.5, 2, and 2.5 $A/dm^2$ and stirring speed is set to 320 r/min in the base bath containing 0.005 g/L SDS to observe the influence of the current density on the size of the bell-mouthed macropores in the deposition. To keep the thickness of the deposition layer approximately consistent, different deposition times are adopted under different current density conditions.

Figure 6a–d shows the frontal morphology of nickel deposition at a current density of 1, 1.5, 2, and 2.5 $A/dm^2$. In addition to bell-mouthed macropores in the deposition,

there are a few pinhole defects. Figure 6(a1–a4,b1–b4,c1–c4,d1–d4) show the cross-section morphologies of bell-mouthed macropores at current densities of 1, 1.5, 2, and 2.5 A/dm$^2$, respectively. Similarly, the extended contour line measures the opening angle of the bell-mouthed macropores and the angle values are listed in Table 3. As the solution current density increases from 1 to 2.5 A/dm$^2$, the angle range of the bell-mouthed macropore cross-sectional profile is increased from 39.3° to 46.3°. When the current density is 2 and 2.5 A/dm$^2$, the contour angle of the bell-mouthed macropores is not much different but the angles ranging under 2.5 A/dm$^2$ are more concentrated. The increase in current density makes the deposition "climb" along the bubbles more quickly and thus reduces the portion of the bubbles outside the deposition surface faster, which will reduce the contact angle between the bubbles and the surface of the deposition, so the size of the bell-mouthed macropores decreases. However, the hydrogen evolution reaction is strengthened due to the increase in the current density increasing cathode polarization, which promotes the growth of bubbles attached to the deposition. The effect of an increase in the current density promotes the growth of bubbles more than the acceleration of the deposition growth reducing the contact angle of bubbles. As a result, the sizes of the bell-mouthed macropores increase.

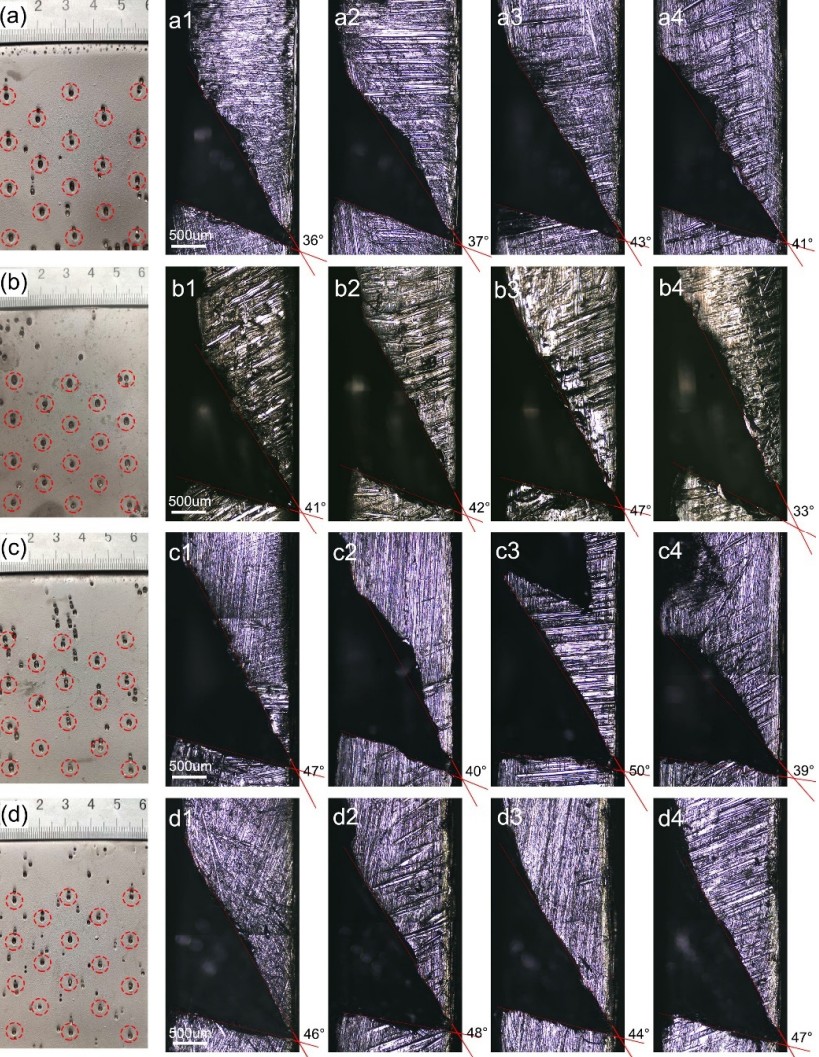

**Figure 6.** The morphology of bell-mouthed macropores in a deposition under different current densities: (**a**) 1 A/dm$^2$, 180 h; (**b**) 1.5 A/dm$^2$, 120 h; (**c**) 2 A/dm$^2$, 90 h; (**d**), 2.5 A/dm$^2$, 72 h. (**a**–**d**) and (**a1**–**d4**) are the front and cross-sectional morphology of the macropore, respectively.

**Table 3.** Cross-section angles of bell-mouthed macropores in a deposition at different current densities.

| Current Density (A/dm$^2$) | 1 | 1.5 | 2 | 2.5 |
|---|---|---|---|---|
| Average angle of macropores profile (°) | $39.2 \pm 3.3$ | $39.5 \pm 4.3$ | $44 \pm 5.4$ | $46.3 \pm 1.7$ |

## 4. Conclusions

In conclusion, in this work, a nickel layer with bell-mouthed macropores was prepared by using a substrate with micropores and hydrogen bubbles as templates. Based on our results, salient conclusions can be drawn as follows:

- The surface tension of the solution is a decisive factor in obtaining the target nickel layer. When the content of SDS in the base bath is 0.0050 g/L ($\gamma_L$ = 50.2 mN/m), the desired bubble template can be formed and there are fewer bubbles attached to other areas on the substrate. The bubble template is relatively static due to the hydrogen bubbles staying attached to the surface of the substrate, and the bubbles grow slowly. The micropores on the substrate provide a preferential condition for the formation of the bell-mouthed macropores in the deposition and make the positions of the bell-mouthed macropores controllable.
- The sizes of the bell-mouthed macropores can be adjusted by varying the stirring speed of the solution and the current density. As the solution stirring speed increases from 160 to 480 r/min, the angle range of the bell-mouthed macropore cross-sectional profile is increased from 21.0° to 44.3°. In addition, the angle range of the bell-mouthed macropores cross-sectional profile is increased from 39.3° to 46.3° with the current density increasing from 1 to 2.5 A/dm$^2$.

The macroporous nickel prepared in the study can be used to manufacture an electroforming mold with a vacuum forming function, which can improve the production efficiency of traditional electroforming molds.

**Author Contributions:** Conceptualization, R.Y. and W.Y.; methodology, R.Y. and G.Q.; validation, R.Y. and W.Y.; investigation, R.Y.; resources, Y.D.; data curation, R.Y.; writing—original draft preparation, R.Y.; writing—review and editing, Y.D.; visualization, Y.D.; supervision, Y.D.; project administration, Y.D.; funding acquisition, Y.D. All authors have read and agreed to the published version of the manuscript.

**Funding:** This work is supported by the Science and Technology Development Project of Jilin Province, China (no. 20190302092GX).

**Data Availability Statement:** Not applicable.

**Conflicts of Interest:** The authors declare no conflict of interest.

## Abbreviations

The list of nomenclature and abbreviations for all the symbols and Greek letters reported in the article.

| | |
|---|---|
| $w_{ad}$ | Unit adhesion energy |
| $E_{ad}$ | Adhesion energy; the work required in detaching the bubble from the electrode |
| $S_{air}$ | Area of interface between bubble and deposition |
| $\gamma_L$ | Surface tension of the H$_2$–liquid interface |
| $\theta_{H2}$ | Contact angle of the bubble at the cathode surface |
| DHBT | Dynamic hydrogen bubble template |
| SDS | Sodium dodecyl sulfate |

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
