# Peer review of "Preparation of a Nickel Layer with Bell-Mouthed Macropores via the Dual-Template Method"

_metals, doi:10.3390/met11121894_

Round 1

Reviewer 1 Report

Thank you for submitting your paper. The work done here draws attention to a significant subject in depositing nickel material. I have found the paper to be interesting. However, several issues need to be addressed properly before the paper is being considered for publication. My comments including major and minor concerns are given below:

  1. Please consider reviewing the abstract and highlight the novelty, major findings, and conclusions. I suggest reorganizing the abstract, highlighting the novelties introduced. The abstract is too short and should be expanded so that it contains answers to the following questions:
  2. What problem was studied and why is it important?
  3. What methods were used?
  4. What conclusions can be drawn from the results? (Please provide specific results and not generic ones) for example effect of pores and stirring speed. Please use numbers or % terms to clearly shows us the results in your experimental work.
  5. What is the novelty of the work and where does it go beyond previous efforts in the literature?
  6. Just before the last paragraph in the introduction, the authors should answer the following question: What is the research gap did you find from the previous researchers in your field? Mention it properly. It will improve the strength of the article.
  7. The introduction part is too short and should be expanded upon. The literature review is basic and generic about how processes occur, it does not provide the readers with a clear understand of the problem in hand. Please use more in-depth critical review.
  8. Also, please cite papers related to your work from mdpi journals.
  9. Materials and methods section is comprehensive and clear, however, images and graphs of equipment used, samples fabricated, and tests implemented with details on those images should be provided, this is an experimental study, and it is important to give sufficient information to the readers about the work done here.
  10. In the materials and method section, please indicate if any standards were used to perform the substrate preparation process.
  11. Please combine all small paragraphs into larger ones. For example, lines 97-107 combine into one larger paragraph, please check this issue elsewhere in the manuscript.
  12. Please add a list of nomenclature and abbreviations for all the symbols and Greek letters reported in the article at the end of the manuscript.
  13. Line 119 “while the bell-mouthed macropores at the micropores formed” please revise this sentence.
  14. Extensive editing of English language and style required
  15. Line 127 “the micropores due to the content of SDS increase more than 0.0060g/L” this sentence needs rephrasing. The English writing style and grammar of the article should be thoroughly checked.
  16. Line 128-129 how about past studies, did they find similar results to yours or different, in either way please discuss and support with references if possible.
  17. Line 152-153 “which appear upward duo to the bubbles appear upward under the effect of buoyancy.” This sentence is not clear and does not provide a useful information/meaning.
  18. Line 162 please support this claim with references.
  19. Line 165-167 “The flow velocity of the electrolyte on the cathode surface is directly determined by the stirring speed of the solution.” Please support this claim with references, also what did past studies find, does it agree or disagree with your finding?
  20. Lines 195-198 please support this claim with a reference(s).
  21. Line 218 please rephrase the sentence it does not read well.
  22. The results are merely described and is limited to comparing the experimental observation. The authors are encouraged to include a more detailed results and discussion section and critically discuss the observations from this investigation with existing literature.
  23. The authors show section 3. Results but where is the discussion section? Or is this section supposed to be renamed results and discussion?
  24. Conclusion can be expanded or perhaps consider using bullet points (1-2 bullet points) from each of the subsections.

Reviewer 2 Report

The authors describe a methodology to “guide” the formation of bubbles during Ni electrodeposition. They use these bubbles as a template for the realization of bell shaped macropores. The concept itself is interesting, but its investigation in the paper is quite incomplete. The authors state that their macropores can be generically used as molds for vacuum forming, but they do not demonstrate this possibility. Consequently, I assume that this is a fundamental research paper that describes the methodology in general. If this is true, however, the level of the characterization carried out must be high and the results should be properly discussed. Some observations:

1) The paper looks quite “artisanal”. No advanced characterization techniques are employed. For example, it is important to investigate the internal morphology of the pores as a function of deposition parameters by SEM.

2) If this is a fundamental paper, the effect of some important parameters has not been investigated. What is the influence of the diameter of the laser drilled micropores? What is the influence of the substrate inclination inside the Ni plating solution? I see that the pores are inclined. I imagine that, if the sample is placed horizontally in the plating solution, the pores should be perfectly vertical and symmetrical.

3) Some method are missing: how was the contact angle calculated? How did the authors prepare the sections? How did they apply the insulating glue?

4) There is a mistake in the caption of figure 3f. It should be 0.0100 g/L instead of 0.0010 g/L.

5) Lines 162: what is the origin of the microbubbles described by the authors?

6) Figure 6 d3 is out of focus.

7) Tables 2 and 3: I suggest to replace ranges with single values and standard deviations. Standard deviations should be provided also for the data reported in table 1.

Regarding the quality of the English used in the manuscript, I’m not fully qualified to judge it. I believe, however, that it should be improved (especially in the introduction). Some phrases are really difficult to understand. Some examples: lines34-35, lines 37-39, lines 49-50, lines 150-153.

Author Response

List of responses to comments

Manuscript ID: metals-1443131

Manuscript name: Study on curing and flammability properties of UV-curable flame-retardant coating on jute/polypropylene composites surface

Dear Reviewer,

Thank you very much for your help in processing the review of our manuscript (Manuscript ID: metals-1443131). We have carefully read these suggestions which are very helpful for us to improve our manuscript. On the basis of enlightening questions and helpful advices, we have now completed the revision of our manuscript. The itemized responses to comments are listed in the succeeding content. We hope that all these corrections and revisions would be satisfactory. Thanks a lot, again.

The specific modifications are as follows:

Comments and Suggestions:

The authors describe a methodology to “guide” the formation of bubbles during Ni electrodeposition. They use these bubbles as a template for the realization of bell shaped macropores. The concept itself is interesting, but its investigation in the paper is quite incomplete. The authors state that their macropores can be generically used as molds for vacuum forming, but they do not demonstrate this possibility. Consequently, I assume that this is a fundamental research paper that describes the methodology in general. If this is true, however, the level of the ch8aracterization carried out must be high and the results should be properly discussed. Some observations:

Reply: We appreciate for your valuable comment.
We are very sorry about “The authors state that their macropores can be generically used as molds for vacuum forming, but they do not demonstrate this possibility”. The following picture will illustrate this kind of application. As showing in the picture, the bell-mouthed through macropores are distributed in the mold, and the small end of the macropores is on the side of the molding surface of the mold. When the polymer leather is put into the molding cavity, a closed cavity can be formed between the leather and the mold. The vacuum equipment vacuums the mold through the large holes, and at the same time softens the polymer leather by heating, then the leather can be formed (See the “In Mold Graining” process for details). The mold in the picture has a vacuum forming function, which is superior to the traditional electroforming mold which only has a thermoforming function. The mold in the picture is mainly used in the production of automotive interior skins, which have complex shapes and fine textures on the surface.

1) The paper looks quite “artisanal”. No advanced characterization techniques are employed. For example, it is important to investigate the internal morphology of the pores as a function of deposition parameters by SEM.

Reply: Thank you for your constructive and helpful suggestion.
The thickness of the nickel layer prepared in this article is about 2 mm, and the largest pore diameter in the nickel layer is about 3-4 mm, so they belong to the macroscopic field and can be observed clearly by using a light microscope, so SEM is not used to observe the morphology of the macropore wall. The wall of the macropore is formed by the deposition of nickel ions along the surface of the bubble. It is only a smooth curved surface, which can be observed in Figures 5b and c. (Line 202, Page 7)

2) If this is a fundamental paper, the effect of some important parameters has not been investigated. What is the influence of the diameter of the laser drilled micropores? What is the influence of the substrate inclination inside the Ni plating solution? I see that the pores are inclined. I imagine that, if the sample is placed horizontally in the plating solution, the pores should be perfectly vertical and symmetrical.

Reply: Thank you for your constructive and helpful suggestion.
The pore size of the micropores on the substrate does have an impact on the experimental results, such as bubble formation and growth rate. This article is discussed under the condition that the pore diameter is determined, and future work may discuss the influence of the micropore diameter.
The inclination of the substrate in the Ni plating solution does affect the pore morphology. We have placed the substrate horizontally at the bottom of the solution, and the macropores obtained in the deposition are completely vertical and symmetrical. However, when the substrate is placed horizontally, the requirements for equipment are higher, and a larger water bath and independent stirring device are required. When the substrate is placed vertically, the device is simple and the electrodeposition process is also easy to maintain. It is important that the inclination of the substrate does not affect the exploration of the mechanism and the influence of parameters.

3) Some methods are missing: how was the contact angle calculated? How did the authors prepare the sections? How did they apply the insulating glue?

Reply: We gratefully appreciate for your comment.
The influence of bubble contact angle in the article is only a general discussion, and no specific value is involved, so the calculation of contact angle is not introduced. The preparation of the sections and the application of insulating glue are added in “2. Materials and Methods” as follows:

“The beaker is placed in a water bath pot (Yuhua, China) which provides heating and stir-ring for the electrodeposition. A DC power supply (Longwei, China) is used to provide power.” (Line 103-105)
“The solution with different contents of SDS was heated to 50°C, and then the surface tension of the solution was quickly tested using a surface tensiometer (Youte, China). The nickel deposition is cut by a cutting machine (Dongcheng, China) in the vertical direction, then use a file to grind the section to the symmetry line of the macropore, and then use 400, 1200 grit sandpaper to grind the section to obtain the section morphology. Digital camera (Canon, Japan) was used to directly observe the surface morphology of the deposition, and the section morphology of the bell-mouthed macropores were collected by optical micro-scope (CNOPTEC, China). In the software Photoshop, the outline of the section morphology of the macropore is extended to intersect, and the measured angle is used to measure the size of the macropore.” (Page 2, Line 106-110)
“The 200um through micropores are fabricated by laser on the copper sheet, then drop in-sulating glue (Kafuter) on the opening of the through micropore on the insulating side of the copper sheet to block the micropore.” (Page 3, Line 113-115)

4) There is a mistake in the caption of figure 3f. It should be 0.0100 g/L instead of 0.0010 g/L.

Reply: Thank you for your helpful suggestion.
It has been modified in Line 153, Page 4.

5) Lines 162: what is the origin of the microbubbles described by the authors?

Reply: Thank you for your constructive and helpful comment.
The pH of the electroplating solution used in this article is around 4, and the solution contains a large amount of hydrogen ions. During the electrodeposition process, hydrogen ions will be reduced on the surface of the cathode to generate hydrogen micro-bubbles, and these hydrogen micro-bubbles are dissolved in the solution after desorption. It is mentioned in Line 159-160 and the experimental phenomenon in Figure 2 (Line 151, Page 4) also proves it.

6) Figure 6 d3 is out of focus.

Reply: Thank you for your constructive and helpful comment.
It has been replaced in Line 200, Page 7.

7) Tables 2 and 3: I suggest to replace ranges with single values and standard deviations. Standard deviations should be provided also for the data reported in table 1.

Reply: Thank you for your valuable suggestion.
It has been modified in Line 157, Page 5; Line 221, Page 8; Line 249, Page 9.

Regarding the quality of the English used in the manuscript, I’m not fully qualified to judge it. I believe, however, that it should be improved (especially in the introduction). Some phrases are really difficult to understand. Some examples: lines34-35, lines 37-39, lines 49-50, lines 150-153.

Reply: We appreciate for your valuable comment.
These sentences have been revised as follows: “When the bubbles appear a longer residence time, the larger pores in deposition are obtained. And the smaller pores form in deposition with the coalescence of bubbles is suppressed.” (Line 41-42, Page 1)

“For this phenomenon, the bubble template behavior proposed by Liu [22] is typical, in which the bubbles become more and more larger as the bubble evolution.” (Line 45-46, Page 2)

“It indicates that the bubbles on the cathode stay attached and gradually grow up, and the behavior of this bubble template is relatively static and completely different from DHBT, which bubbles generate and desorb very quickly.” (Line 58-60, Page 2)

“Because the deposition deposits along the surface of the bubbles template, when the con-tact angle between the bubbles and the surface of the deposition is an obtuse angle, a bell-mouthed macropore will be formed in the deposition. In addition, due to the buoyan-cy of the bubbles, the bell-mouthed macropores in the deposition are inclined upward.” (Line 1711-175, Page 5)

Thank you for your serious and constructive comments on our manuscript, again.

Sincerely,

Yanli Dou,

College of Materials Science and Engineering,

Key Laboratory of Automobile Materials of Ministry of Education.

Jilin University,

Changchun, Jilin, 130025, China

2021-11-11

Round 2

Reviewer 1 Report

Dear Authors, please send me the revised version with highlights in yellow for all the changes made, i cant see them in the current new version.

Also please move the table at the end to after the conclusions section.

Reviewer 2 Report

Regarding point number 2, the authors replied:

"The pore size of the micropores on the substrate does have an impact on the experimental results, such as bubble formation and growth rate. This article is discussed under the condition that the pore diameter is determined, and future work may discuss the influence of the micropore diameter".

I can understand that the authors want to investigate the influence of pore size in a future work. However, they should at least discuss here the expectable effect of changing the pore size. In my opinion, it is fundamental to make the reader aware of this possible parameter.

"The inclination of the substrate in the Ni plating solution does affect the pore morphology. We have placed the substrate horizontally at the bottom of the solution, and the macropores obtained in the deposition are completely vertical and symmetrical. However, when the substrate is placed horizontally, the requirements for equipment are higher, and a larger water bath and independent stirring device are required. When the substrate is placed vertically, the device is simple and the electrodeposition process is also easy to maintain. It is important that the inclination of the substrate does not affect the exploration of the mechanism and the influence of parameters.

It's nice to hear that the authors already tried to place the substrate horizontally. I invite them to enclose these data in the manuscript. Controlling the inclination of the pores is a fundamental part of the work in view of the final vacuum molding application.

Round 3

Reviewer 1 Report

Thank you, all questions were answred to satisfactory level and paper can be accepted